# Modulation of FDG Uptake by Cell Cycle Synchronization Using a T-Type Calcium Channel Inhibitor

**DOI:** 10.3390/cancers15215244

**Published:** 2023-10-31

**Authors:** Joon-Kee Yoon, Won Jun Kang

**Affiliations:** 1Department of Nuclear Medicine and Molecular Imaging, Ajou University School of Medicine, Suwon 16499, Republic of Korea; jkyoon3@ajou.ac.kr; 2Department of Nuclear Medicine, Yonsei University College of Medicine, Seoul 03722, Republic of Korea

**Keywords:** FDG PET, cell cycle synchronization, G1 arrest, T-type calcium channel inhibitor, mibefradil

## Abstract

**Simple Summary:**

Cell cycle synchronization method could be used to improve the diagnostic sensitivity of oncologic FDG positron emission tomography.

**Abstract:**

Background: We investigated whether cell cycle synchronization induced by the T-type calcium channel inhibitor mibefradil could increase tumoral 2-[^18^F] fluoro-2-deoxy-d-glucose (FDG) uptake in vitro and in vivo. Methods: Human prostate cancer cells (PC-3) were treated with 10 μM mibefradil for 24, 48, and 72 h to induce G1 arrest. Cell cycle distribution was analyzed at 0, 4, 8, 12, 15, 18, and 24 h after mibefradil withdrawal. Cellular uptake was measured after incubating cells with [^3^H] Deoxy-d-Glucose (DDG) for 1 h at the same time points used in the cell cycle analysis. The correlation between [^3^H] DDG uptake and each cell cycle phase was evaluated in the early (0–12 h) and late phases (15–24 h) of synchronization. In vivo FDG PET imaging was performed in PC-3-bearing mice at baseline, 24 h, and 48 h after mibefradil treatment. Results: The G0/G1 fraction of PC-3 cells was significantly increased from 33.1% ± 0.2% to 60.9% ± 0.8% after 24 h mibefradil treatment, whereas the S and G2/M fractions were decreased from 36.3% ± 1.4% to 23.2% ± 1.1% and from 29.7% ± 1.3% to 14.9% ± 0.9%, respectively, which were similar to the results by serum starvation. Mibefradil treatment for 24, 48, and 72 h increased the number of cells in S phase at 18–24 h after withdrawal; however, only the 72 h treatment increased [^3^H] DDG uptake (145.8 ± 5.8% of control at 24 h after withdrawal). [^3^H] DDG uptake was positively correlated with the size of the S phase fraction and negatively correlated with the size of the G0/G1 fraction in the late phase of synchronization. DDG uptake was significantly increased by mibefradil-induced cell cycle synchronization and correlated with the sizes of cell cycle fractions. In vivo FDG PET imaging also demonstrated a significant increase in tumor uptake after mibefradil treatment. Quantified tumor FDG uptake (%ID/g) increased from 4.13 ± 2.10 to 4.7 ± 2.16 at 24 h, and 5.95 ± 2.57 at 48 h (*p* < 0.05). Conclusion: Cell cycle synchronization could be used to increase the diagnostic sensitivity of clinical FDG positron emission tomography.

## 1. Introduction

Positron emission tomography (PET) with 2-[^18^F] fluoro-2-deoxy-d-glucose (FDG) has been widely used in oncology in the evaluation of a variety of cancers [1], as most tumor cells have a significantly higher rate of glucose consumption than normal cells according to Warburg [2]. Intracellular FDG accumulation is dependent on glucose transporters (GLUTs) and hexokinase, which actively transports FDG into cells and phosphorylates it to FDG-6-phosphate. In many tumors, high FDG uptake is closely connected with the increased expression of GLUT and hexokinase [3,4,5]. Conversely, a few tumors exhibit low FDG uptake, which often reduces the diagnostic sensitivity of FDG PET [6,7,8,9]. For example, FDG PET has moderate sensitivity to prostate cancer because of the low glucose utilization of well-differentiated cancer, and thus, it is difficult to differentiate prostate cancer, benign prostate hyperplasia, and prostatitis according to FDG uptake [6]. Similarly, the sensitivity of FDG PET in detecting hepatocellular carcinoma is modest because of the low expression of GLUT-2 and relatively high activity of glucose-6-phosphatase in well-differentiated cancers [7,8,9]. Gastric signet ring cell carcinoma also has low tumoral FDG uptake owing to the low level of membranous GLUT-1 expression [10]. The superiority of other radiopharmaceuticals over FDG in evaluating these tumors has been demonstrated. As prostate-specific membrane antigen (PSMA) is overexpressed in most prostate cancers, radiolabeled PSMA shows higher diagnostic sensitivity in detecting lymph node and distant metastasis, recurrences, and intraoperative tumor localization than other imaging modalities [11]. A few radiotracers that perform better than ^18^F-FDG have been documented in hepatocellular carcinoma. Of these, acetate, a precursor of phosphatidylcholine, exhibits high sensitivity, especially for well-differentiated tumors [12]. ^11^C-acetate and ^18^F-FDG have complementary roles in the detection of hepatocellular carcinoma. In a previous study with gastric signet ring cell carcinoma, radiolabeled fibroblast-activating protein inhibitors showed higher radiotracer uptake than ^18^F-FDG and thus were more sensitive for detecting the primary tumor, lymph node metastasis, and local recurrence [13]; however, these radiopharmaceuticals cannot replace FDG because of the wide availability of FDG and the ability of FDG to detect other malignancies [7,14]. Therefore, these specific radiopharmaceuticals can only be used to supplement FDG, which would make the assessment process more time-consuming and costly. Consequently, the sensitivity of FDG PET itself must be improved. Our hypothesis is that enhancing the sensitivity of ^18^F-FDG in these tumors through a simple preparation would be better than using multiple diagnostic tests.

As most cancer cells proliferate constantly while replicating DNA and dividing into daughter cells, they pass through a series of cell cycle phases. A variety of cell-cycle-based agents have been applied to cancer treatment as single agents or in combination with conventional chemotherapy [15]. Among these agents, T-type calcium (Ca^2+^) channel inhibitors are promising, and their efficacy as cancer-targeted therapies was previously validated [16,17]. T-type Ca^2+^ channels increase during the G1-S transition, and they play an important role in the progression of the cell cycle by allowing the entry of extracellular Ca^2+^. T-type Ca^2+^ inhibitors serve as negative regulators that block Ca^2+^ entry, which leads to the G1-S cell cycle arrest [18]. Accordingly, the blockade of Ca^2+^ entry would increase the sensitivity of cancer cells to chemotherapeutics such as alkylating agents that induce DNA damage by preventing cell cycle progression [19,20,21]. T-type Ca^2+^ channels can be found in nervous, endocrine, and cardiovascular systems in humans. However, compared with healthy cells, T-type Ca^2+^ channels are more expressed in various cancer cells. In addition, cancer cells can proliferate with lower Ca^2+^ concentration and more rapidly than normal cells. Therefore, cancer cells are more susceptible to the T-type Ca^2+^ channel inhibitors than healthy cells [16,22,23].

The potential of one T-type Ca^2+^ channel inhibitor, mibefradil, to increase survival in combination with temozolomide in an animal model of glioblastoma multiforme was demonstrated [20]. The strategy of that study is displayed in Figure 1. First, when the tumor cells were treated with mibefradil (“mibefradil on”), they were synchronized in G1 because the blockage of Ca^2+^ influx prevented progression beyond the G1/S checkpoint. Subsequently, after mibefradil was withdrawn (“mibefradil off”), synchronized cells re-entered the cycle and progressed from G1 to S phase, which increased the size of the S phase fraction. Tumor cells in S phase are more responsive to chemotherapy.

Although it has not been properly evaluated whether tumoral FDG uptake is related to specific cell cycle phases, several studies indicated that FDG uptake is highly correlated with the proliferative potential of malignant tumors [24,25]. Moreover, it has been reported that lymphomas with large S phase fractions exhibit high FDG accumulation [26]. Based on these reports, we investigated whether the T-type Ca^2+^ channel inhibitor mibefradil could increase tumoral FDG uptake in vitro and in vivo via cell cycle synchronization, and if this hypothesis is true, determine whether FDG uptake is related to specific cell cycle fractions.

## 2. Materials and Methods

### 2.1. Cell Lines and Cell Culture

Human prostate adenocarcinoma (PC-3), human neuroblastoma (SK-N-SH), human breast cancer (MDA-MB-231), and human hepatocellular carcinoma cells (SNU-449) were obtained from the American Type Culture Collection. These cells lines were chosen in consideration of their T-type Ca^2+^ channel expression [23]. MDA-MB-231 and SNU-449 cells were grown in RPMI 1640 medium supplemented with 10% fetal bovine serum and 1% penicillin/streptomycin in a humidified incubator with 5% CO_2_ at 37 °C. PC-3 and SK-N-SH cells were grown in F-12K medium and Earle’s minimum essential medium, respectively, supplemented with the same amount of FBS and antibiotics. The culture medium was changed every 3 days, and cells were subcultured twice a week. After pilot experiments, cells at passage 4 to 12 after recovery were used in this study for the reproducibility of cellular FDG uptake.

### 2.2. Mibefradil Treatment and Serum Starvation

Mibefradil (Tau Therapeutics, LLC, Charlottesville, VA, USA) was dissolved in distilled water and stored at −70 °C. Before cell cycle analysis, the maximum dose that did not provoke morphological changes or weaken cell attachment was designated the optimal dose (5 μM for SK-N-SH cells and 10 μM for PC-3, MDA-MB-231, and SNU-449 cells).

To verify the cell cycle synchronization induced by mibefradil, another group of cells was grown in serum-starved medium for 24 h, and the cell cycle distribution of these cells was compared to that of cells treated with mibefradil.

### 2.3. Cell Cycle Analysis

For cell cycle analysis, 5 × 10^4^ (PC-3, MDA-MB-231, and SNU-449) or 1 × 10^5^ (SK-N-SH) cells were seeded on 12-well plates and grown before mibefradil treatment. Mibefradil was added to each well at a final concentration of 5 (SK-N-SH) or 10 μM (PC-3, MDA-MD-231, and SNU-449) and incubated for 1–3 days. Cells were washed twice with phosphate-buffered saline (PBS) and supplied with fresh medium to eliminate the effect of mibefradil. For control cells, all procedures followed those used for treated cells except that vehicle was added instead of mibefradil. Cells were collected by trypsinization at 0, 4, 8, 12, 15, 18, and 24 h after mibefradil withdrawal and rinsed twice with 5 mL of PBS containing 0.1% bovine serum albumin. After centrifugation, cells were fixed by dripping 3 mL of cold 70% ethanol onto the cell pellets and stored in a 4 °C refrigerator until analysis. After overnight fixation, all samples were centrifuged and washed twice with PBS. Then, cells were resuspended in propidium iodide (PI) staining solution (0.1% (*v*/*v*) Triton X-100, 10 μg/mL PI, and 100 μg/mL DNAse-free RNAse A in PBS) and incubated at room temperature for 30 min in the dark. Cellular fluorescence cells were measured using the flow cytometer (BD FACSCalibur™, San Jose, CA, USA). Data were analyzed using Flowjo software (ver 7.2.4, Tree Star Inc., Ashland, OR, USA).

The confluency of treated cells was approximately 70% when G1 arrest was induced. This experiment was performed in triplicate for all cell lines excluding SK-N-SH cells (n = 2 for treated cells).

### 2.4. Cellular Uptake

Cells were seeded and treated as described for cell cycle analysis. At each time point, cells were washed twice with 1 mL of PBS and incubated with 0.2 μCi [1,2-^3^H(N)]-2-deoxy-d-glucose ([^3^H] DDG, Moravek Biochemicals, Brea, CA, USA) in 1 mL of Hank’s Balanced Salt Solution for 1 h at 5% CO_2_ and 37 °C. Then, cells were washed twice with 2 mL of ice-cold PBS and incubated with 250 μL of 0.1 N NaOH for 10 min at room temperature. Cell lysates (150 μL) were incunated with 10 mL of liquid scintillation fluid (Cytoscint™, Fisher Scientific, Fair Lawn, NJ, USA) for 15 min; then, radioactivity was measured using a liquid scintillation counter (LS 6500, Beckman Coulter Inc., Brea, CA, USA). For the protein assay, 10 μL of cell lysate was incubated with 150 μL of Pierce 660 nm protein assay reagent (Thermo Pierce Scientific Inc., Rockford, IL, USA) for 20 min in 96-well plates (in duplicate). Absorbance was measured using a microplate reader (Synergy 2 multi-mode microplate reader, Bio Tek Instruments Inc., Winooski, VT, USA). Cellular [^3^H] DDG uptake (%uptake/μg) was calculated as the total radioactivity in the cell lysate divided by the radioactivity of 0.2 μCi [^3^H] DDG and the total amount of protein in the cell lysate (μg). This experiment was performed in triplicate.

### 2.5. Animal Model

All animal studies were reviewed and approved by the Institutional Animal Care and Ethics Committee of the Laboratory Animal Research Center and were performed in accordance with the guiding principles of the Care and Use of Laboratory Animals. The reporting in the manuscript follows the recommendations in the ARRIVE guidelines.

Five-week-old female BALB/C nude mice (n = 6, per each group) were injected with a suspension of 1 × 10^6^ PC3 cells subcutaneously in the thigh. A dose of 25 mg/kg of mibefradil was administered orally for 3 days.

### 2.6. In Vivo PET Imaging

PC-3-tumor-bearing mice were intravenously injected with 7.4 ± 0.5 MBq of FDG via the tail vein. In vivo PET imaging was performed using an Inveon microPET/CT scanner (Siemens Medical Solutions USA Inc., Knoxville, TN, USA). FDG PET imaging was acquired at baseline, at 24 h, and 48 h after 3-day mibefradil treatment. Static FDG PET imaging was acquired at 60 min after FDG injection. Each static imaging study was performed for 10 min. Semi-quantification of PET images was performed based on region-of-interest (ROI) analysis, using AMIDE software (SourceForge, New York, NY, USA). A 3D ROI was drawn around the tumor by visual inspection. The maximum activities were recorded from the whole ROI. Uptake values were quantified into maximum %ID/g in the tumor. Images were presented using color scale proportional to tissue concentration.

### 2.7. Statistical Analysis

Data are presented as the mean ± standard error. T-tests were performed to compare the cell cycle distribution and cellular uptake between the control and mibefradil-treated groups. Spearman’s rank correlation was used to assess the relationship between cellular uptake and individual cell cycle fractions. *p*-values less than 0.05 were considered significant.

## 3. Results

### 3.1. Cell Cycle Synchronization Induced by Mibefradil

We tested four cell lines: PC-3, SK-N-SH, MDA-MD-231, and SNU-449 cells. Figure 2A shows the cell cycle distribution after mibefradil treatment. Immediately after 10 μM mibefradil treatment for 24 h, the G0/G1 fraction of PC-3 cells increased from 33.1% ± 0.2% to 60.9% ± 0.8% (*p* < 0.001), whereas the S and G2/M fractions decreased from 36.3% ± 1.4% to 23.2% ± 1.1% (*p* = 0.002) and from 29.7% ± 1.3% to 14.9% ± 0.9% (*p* = 0.002), respectively. MDA-MD-231 cells exhibited a similar change in cell cycle distribution to PC-3 cells. Whereas the G0/G1 fraction of MDA-MD-231 cells increased from 31.5% ± 1.5% to 59.4% ± 6.3% (*p* = 0.019), the S and G2/M fractions decreased from 37.7% ± 0.7% to 25.5% ± 4.9% (*p* = 0.088) and from 30.9% ± 0.7% to 14.8% ± 1.8% (*p* = 0.001), respectively. Similarly, in SK-N-SH cells, 5 μM mibefradil treatment for 24 h increased the G0/G1 fraction from 48.7% ± 1.5% to 60.7% ± 0.9% (*p* = 0.007) and decreased the S fraction from 39.1% ± 3.6% to 19.2% ± 0.8% (*p* = 0.026). However, this treatment also increased the G2/M fraction from 11.2% ± 2.8% to 21.4% ± 0.1% (*p* = 0.067), which did not reach statistical significance.

On the other hand, SNU-449 cells displayed only a slight increase in the G0/G1 fraction (from 30.6% ± 0.6% to 35.5% ± 0.4%, *p* = 0.005) without significant changes in the S (from 25.3% ± 2.1% to 24.1% ± 2.3%, *p* = 0.734) and G2/M (from 39.7% ± 2.1% to 36.8% ± 2.1%) fractions.

Although mibefradil induced G1 arrest in both PC-3 and MDA-MB-231 cells, PC-3 cells were chosen for further experiments because prostate cancers show low FDG uptake in clinical PET imaging. Moreover, the decrease in the S fraction of MDA-MB-231 was not statistically significant.

### 3.2. Comparison of the Cell Cycle Distribution between Serum Starvation and Mibefradil Treatment

To verify the induction of G1 arrest by mibefradil, the cell cycle distribution was compared between mibefradil-treated and serum-starved PC3 cells (Figure 2B). After 24 h serum starvation, the G1 fraction was increased by 13.2% ± 0.3% (*p* < 0.001), whereas the S and G2/M fractions were decreased by 6.2% ± 0.4% (*p* = 0.020) and by 7.2% ± 0.7% (*p* = 0.021), respectively, compared with the control findings. Similar changes were observed after 10 μM mibefradil treatment for 24 h, as the G1 fraction was increased by 18.2% ± 0.6% (*p* = 0.001), whereas the S and G2/M fractions were decreased by 10.5% ± 0.2% (*p* = 0.003) and 7.5% ± 1.2% (*p* = 0.031), respectively.

### 3.3. Cell Cycle Distribution and Cellular [^3^H] DDG Uptake in PC-3 Cells after Mibefradil Treatment

To determine whether cellular FDG uptake is increased by cell cycle synchronization, we investigated the changes in cell cycle distribution and cellular [^3^H] DDG uptake for 24 h after removing mibefradil (Figure 3A–D). When PC-3 cells were treated with 10 μM mibefradil for 24 h, the G0/G1 fraction of treated cells was larger than that of the control cells (59.8% ± 0.8% vs. 31.7% ± 0.5% at 0 h) (Figure 3A); however, the G0/G1 fraction decreased over time after mibefradil withdrawal, and it was smaller than that of the control cells at 24 h (30.2% ± 0.4% vs. 35.3% ± 0.9%). Conversely, the S fraction was initially smaller in treated cells than in control cells (19.7% ± 1.4% vs 38.6% ± 0.6%) but increased at 18 (44.1% ± 0.9% vs. 36.7% ± 1.5%) and 24 h (41.8% ± 1.3% vs. 30.5% ± 3.1%) after withdrawal. However, despite the increase in the S fraction, [^3^H] DDG uptake by treated cells did not increase significantly (76.8–113.5% of controls, all *p* > 0.05) (Figure 3D).

We also investigated the cell cycle distribution and cellular [^3^H] DDG uptake after 48 h with 10 μM mibefradil treatment (Figure 3B). Similar to the findings after 24 h of treatment, the G0/G1 fraction of treated cells was smaller than that of the control at 15–24 h (39.3–44.8% vs. 46.6–47.6%), whereas the S fraction of treated cells was larger than that of the control at 18 (30.1% ± 3.4% vs. 23.3% ± 1.9%) and 24 h (27.0% ± 0.9% vs. 20.5% ± 2.8%) after mibefradil withdrawal. Despite the enlargement of the S fraction, [^3^H] DDG uptake by treated cells was not significantly higher than that by control cells (91.4–124.5%, all *p* > 0.05) (Figure 3D).

However, unlike 24 and 48 h treatment, 72 h mibefradil treatment resulted in a significant increase in [^3^H] DDG uptake at 15–24 h after mibefradil withdrawal (Figure 3D). Compared with control cells, the [^3^H] DDG uptake by treated cells was 118.1% at 15 h (*p* = 0.030), 128.5% at 18 h (*p* = 0.021), and 140.1% at 24 h (*p* = 0.001). In the cell cycle analysis, the S fraction of treated cells was larger than that of control cells at 18 (35.3% ± 1.1% vs. 25.2% ± 0.7%) and 24 h (34.5% ± 0.9% vs. 25.7% ± 1.3%), whereas the G0/G1 fraction was smaller at 18 (45.7% ± 0.5% vs. 55.2% ± 0.7%) and 24 h (42.2% ± 1.0% vs. 57.1% ± 0.7%) (Figure 3C).

Taken together, although 24, 48, and 72 h mibefradil treatments induced G1 arrest, only the 72 h treatment increased [^3^H] DDG uptake significantly.

As the passage of cells may influence the results, we repeated this experiment at three different passages (passages 5, 6, and 8, Table 1). The [^3^H] DDG uptake of treated cells ranged from 91.3 ± 12.0% to 145.8 ± 5.8% of controls. At passage 5, [^3^H] DDG uptake was significantly higher at 18 (134.6% of control, *p* = 0.012) and 24 h (135.1% of control, *p* = 0.001) after drug removal. Similarly, at passage 8, [^3^H] DDG uptake was significantly higher at 18 (114.0% of control, *p* = 0.048) and 24 h (155.1% of control, *p* = 0.047) after drug removal. By contrast, at passage 6, [^3^H] DDG uptake was significantly higher only at 24 h (147.2% of control, *p* = 0.013) after drug removal. Thus, [^3^H] DDG uptake was consistently higher in treated cells than in control cells only at 24 h after mibefradil withdrawal.

### 3.4. Correlation between Cell Cycle Distribution and Cellular [^3^H] DDG Uptake in PC-3 Cells after Mibefradil Treatment

To assess the relationship between individual cell cycle fractions and cellular [^3^H] DDG uptake, control and treated cells were combined according to their respective cell cycle phase and then divided into early (0–12 h) and late phases (15–24 h) of cell cycle synchronization. In the early phase, [^3^H] DDG uptake displayed no correlation with any of the cell cycle fractions (Figure 4A), whereas in the late phase, it showed a positive correlation with the S fraction (r^2^ = 0.861, *p* = 0.008) and a negative correlation with the G0/G1 fraction (r^2^ = 0.659, *p* = 0.049, Figure 4B).

### 3.5. In Vivo PET Imaging after Mibefradil Treatment

In the control group, the baseline tumor uptake (%ID/g) was 4.0 ± 0.5, and measured to be 4.6 ± 1.14 at 24 h, and 4.45 ± 1.24 at 48 h (*p*-value > 0.05) (Figure 5A). No significant change of uptake value was noted.

In the treatment group, tumor uptake (%ID/g) increased from 4.13 ± 2.10 to 4.7 ± 2.16 at 24 h, and 5.95 ± 2.57 at 48 h. The difference between baseline to 48 h and 24 h to 48 h achieved statistical significance (*p* < 0.05) (Figure 5).

## 4. Discussion

The sensitivity of FDG PET is limited in some tumors for reasons related to low FDG accumulation in tumor cells. The best method of overcoming this weakness is to increase FDG uptake in those tumors. In the present study, we investigated cell cycle synchronization induced by the T-type Ca^2+^ channel inhibitor mibefradil for that purpose. As a result, we observed that mibefradil treatment synchronized tumor cells in the G1 phase, and FDG uptake was increased after drug removal both in vitro and in vivo. Additionally, FDG uptake was correlated positively with the S fraction and negatively with the G0/G1 fraction in the late phase of synchronization. To our knowledge, this is the first report of a successful increase in tumoral FDG uptake using cell cycle modulation by T-type Ca^2+^ channel inhibitor.

Mibefradil is a Ca^2+^ channel inhibitor that blocks both T-type and L-type Ca^2+^ channels. Mibefradil was effectively used in the management of hypertension and chronic stable angina; nevertheless, it was withdrawn from the market because of serious drug–drug interactions with β-blockers, digoxin, verapamil, and diltiazem [27]. However, a few years later, mibefradil was revisited because of the ability of T-type Ca^2+^ channel inhibitors to control cell cycle progression and inhibit tumor cell proliferation [22]. As a single agent, mibefradil inhibited the growth of retinoblastoma and breast cancer cells by blocking Ca^2+^ currents and induced necrosis in those cells [28]. Similarly, mibefradil prolonged the survival of mice bearing xenografts from various glioblastoma multiforme cells by 13–50% [20]. In addition, the addition of temozolomide to mibefradil increased the survival benefit by up to 71.9% [20]. This synergistic effect results from the fact that tumor cells synchronized at the G1/S checkpoint enter S phase at the same time, and resultantly, cells become more susceptible to S-phase-specific agents. This increase in the S fraction results in both enhanced sensitivity to chemotherapy and the activation of tumor metabolism. In that sense, glycolysis, the main energy-producing mechanism of tumor cells, could be upregulated by cell cycle synchronization, thus enhancing FDG uptake.

We found that mibefradil-induced cell cycle synchronization increased [^3^H] DDG uptake significantly only after 72 h of treatment. Although the pattern of change in the cell cycle distribution and the respective sizes of the cell cycle fractions over time were similar with different treatment durations, increased [^3^H] DDG uptake was not achieved by 24 h or 48 h treatment. When we repeated the same experiment for validation, the result corroborated that 72 h mibefradil treatment significantly increased [^3^H] DDG uptake by PC-3 cells. Although the degree of increase varied with the cell passage number, [^3^H] DDG uptake at 24 h after mibefradil removal was consistently higher than that by control cells. This suggests that FDG uptake in clinical PET imaging could be increased by cell cycle synchronization, which would improve the sensitivity of PET in tumors with low FDG uptake.

As has been mentioned, the correlation between FDG uptake and the proliferative capacity of tumor cells has been evaluated in a variety of manners, although the variation in FDG uptake according to the cell cycle phase has rarely been studied. In untreated non-Hodgkin’s lymphoma tissues, there was a significant association between high FDG accumulation (in standardized uptake value) and a large S fraction (r = 0.786, *p* = 0.002) [26], which is in agreement with our observations in the current study that [^3^H] DDG uptake was positively correlated with the size of the S fraction in late-phase synchronization. The cell cycle distribution of tumor cells may be changed by chemotherapy. One of the popular chemotherapeutic drugs, 5-fluorouracil, increased the size of the G1 fraction in breast cancer cells in a time-dependent manner; however, it decreased FDG uptake by 50% after 72 h of treatment [29]. Conversely, paclitaxel increased the S fraction significantly after 72 h of treatment, and FDG uptake was increased approximately 2-fold. These results are also consistent with our observation that [^3^H] DDG uptake was negatively correlated with the G1 fraction and positively correlated with the S fraction. Meanwhile, FDG uptake was also increased by 48 h doxorubicin treatment (33%); however, only the size of the G2/M fraction was increased. In contrast to this result, we did not observe any correlation between FDG uptake and the size of the G2/M fraction.

As shown in Table 1, [^3^H] DDG uptake at 0 and 4 h after 72 h treatment was variable compared to that at the other time points. The standard errors of [^3^H] DDG uptake at 0 and 4 h were 12.0% and 15.8%, respectively. Therefore, we separated cells into early and late phases based on the time after synchronization, and according to our expectation, [^3^H] DDG uptake by PC-3 cells displayed a significant correlation with the S and G0/G1 fractions only in the later phase. This result suggests that PC-3 cells may be unstable early after long-term cell cycle synchronization, and some time may be required to increase FDG uptake in cancer patients in the clinic.

A previous study investigating the effect of flavonoids on the proliferation and cell cycle distribution of cancer cells revealed that flavonoid-treated human breast cancer cells (MDA-MB-435 and MCF-7) and human colon cancer cells (HT-29) returned to their normal cycling within a day of flavonoid removal [30]. This is of great importance in terms of the duration of cell cycle synchronization. Increased FDG uptake would not persist for more than 24 h after mibefradil removal, explaining why we did not evaluate [^3^H] DDG uptake at later time points. In the same manner, although G1 arrest eventually produces a large S fraction in cancer cells, it would not potentiate tumor proliferation.

In vivo PET studies showed a gradual increase in FDG uptake 48 h after mibefradil treatment. In contrast, the control group did not show an interval change in FDG uptake. Although various conditions like tumor growth pattern and anti-tumoral effect can influence FDG uptake in this study, the results illustrated the cell-cycle-modulating drugs can be helpful to enhance the FDG uptake of the tumor, which is especially expected with tumors with low FDG uptake like prostate cancer. To overcome the potential confounding factors that influence tumor FDG uptake including tumor heterogeneity, tumor growth rate, and blood glucose level, further studies will be warranted.

Recently, we have reported that cell cycle synchronization using thiazolidinediones enhanced anticancer effects of 2-DDG [31]. Like mibefradil, troglitazone or pioglitazone induced cell cycle arrest at the G1 phase, and then the withdrawal of these drugs increased 24 h accumulated [^3^H] DDG uptake and membranous glucose transporter-1 expression. In addition, 2-DDG treatment in combination with thiazolidinediones showed significant anticancer effects in colon cancer in vitro and in vivo, while 2-DDG alone showed no significant effect. Although the correlation between the cell cycle phase and ^3^H-DG uptake has not been investigated, these results support the use of mibefradil for enhancing tumoral FDG uptake in cancers.

Dual-time-point (or delayed-time-point) imaging of ^18^F-FDG PET is a well-known effective imaging technique to increase tumoral uptake [32]. Dual-time-point imaging is based on the fact that ^18^F-FDG accumulates more in the tumor at the delayed time points because, unlike inflammatory cells, malignant tumors have low glucose-6-phosphatase activity. Dual-time-point imaging has shown superior or similar diagnostic performances compared with the standard imaging method in various cancers such as lung and breast cancers. However, it also has limitations in that ^18^F-FDG accumulation on the delayed imaging is inconsistent, and some inflammation also has a similar pattern of FDG accumulation to cancers. Therefore, dual-time-point imaging needs to be validated further.

In conclusion, cell cycle synchronization induced by the T-type Ca^2+^ channel inhibitor mibefradil resulted in the increased accumulation of cancer cells in S phase and subsequently increased FDG uptake. Animal studies to verify this outcome showed enhanced FDG uptake after mibefradil treatment. Cell cycle synchronization is expected to enhance the diagnostic sensitivity of FDG PET.

## Figures and Tables

**Figure 1 cancers-15-05244-f001:**
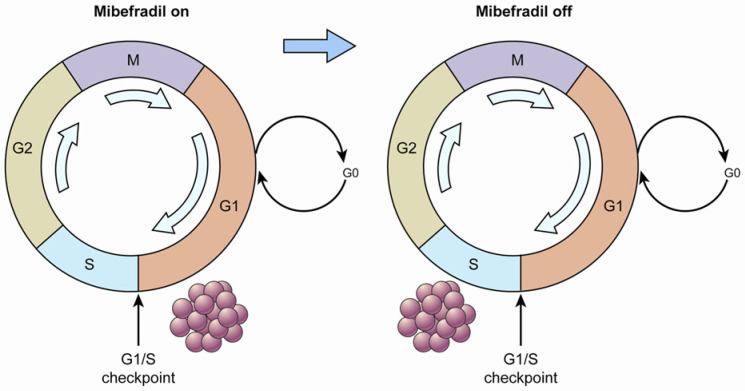
A diagram of cell cycle synchronization. The T-type Ca^2+^ channel mibefradil prevents cells from crossing the G1/S checkpoint. Proliferating cells are synchronized in G1 phase by mibefradil treatment, which results in a larger G1 fraction. After removing mibefradil, cells accumulated in G1 re-enter the cycle, and consequently, the number of cells in S phase increases. Cells in S phase may have a higher metabolic rate than those in the other phases.

**Figure 2 cancers-15-05244-f002:**
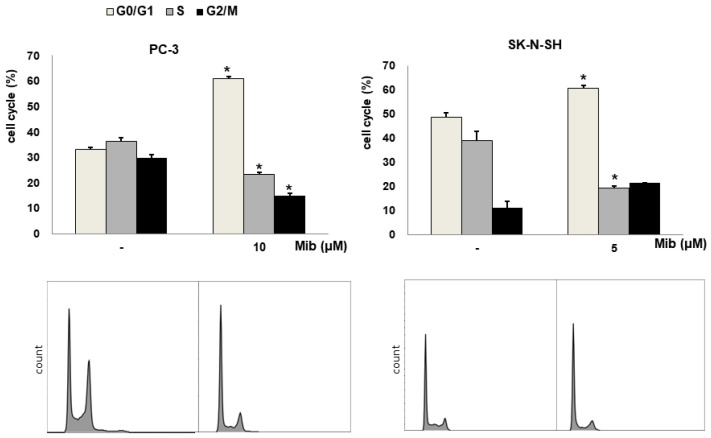
(**A**) Cell cycle distribution changes induced by mibefradil treatment in PC-3 (human prostate adenocarcinoma), SK-N-SH (human neuroblastoma), MDA-MB-231 (human breast cancer), and SNU-449 (human hepatocellular carcinoma) cells. (**B**) Comparison of the cell cycle distribution after serum starvation and mibefradil treatment. The data are presented as the mean and standard error of three independent experiments. * *p* < 0.05. CTRL = control, SS = serum starvation, Mib = mibefradil.

**Figure 3 cancers-15-05244-f003:**
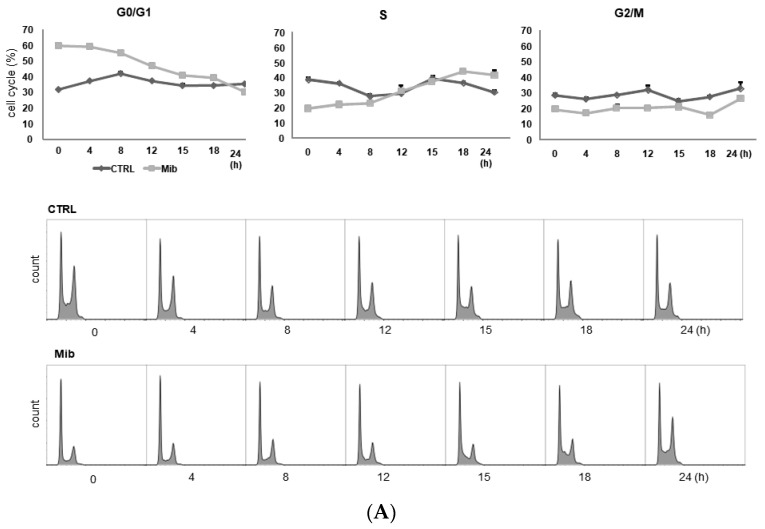
Cell cycle distribution (**A**–**C**) and [^3^H] DDG uptake (**D**) in control and mibefradil-treated PC-3 cells. Cells were incubated with 10 µM mibefradil for 24 h, 48 h, or 72 h. The X-axis indicates time (h) after removing mibefradil. Cellular [^3^H] DDG uptake is presented as the relative percentage of control uptake at 0 h. * *p* < 0.05. CTRL = control, Mib = mibefradil.

**Figure 4 cancers-15-05244-f004:**
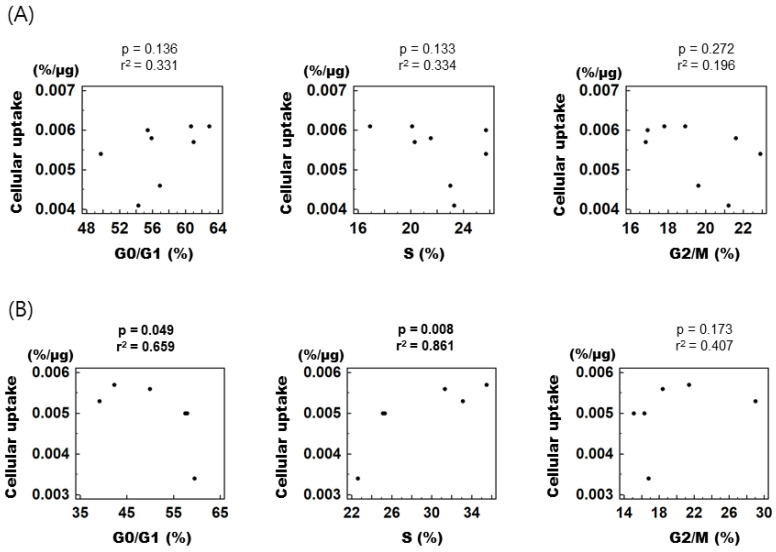
Correlation between specific cell cycle fractions and [^3^H] DDG uptake. (**A**) From 0 to 12 h after mibefradil removal. (**B**) From 15 to 24 h after mibefradil removal. r^2^ indicates Pearson’s correlation coefficient. Cellular [^3^H] DDG uptake is presented as %/μg.

**Figure 5 cancers-15-05244-f005:**
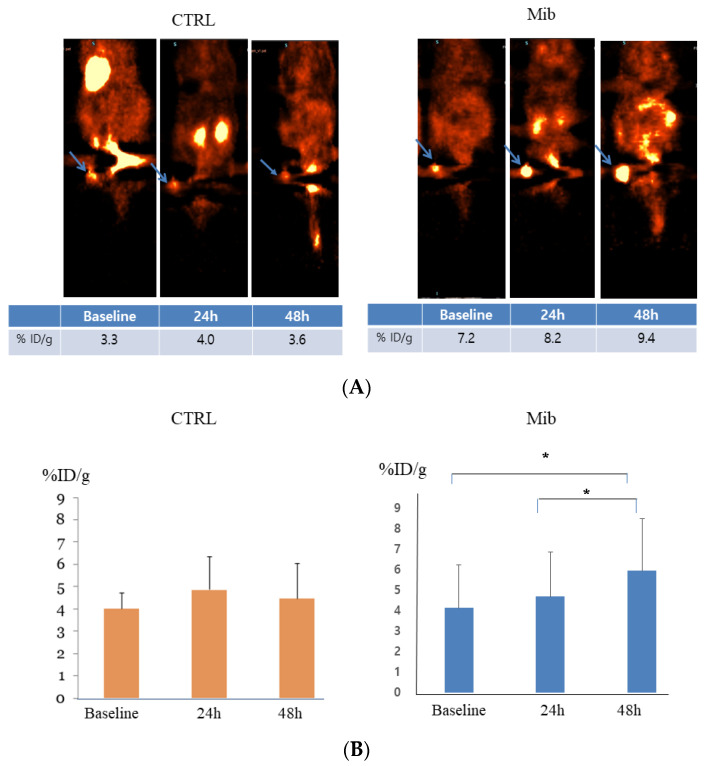
In vivo FDG PET imaging in the control and treatment groups. PC-3-bearing tumor showed no remarkable interval change over time in the control group, while tumor FDG uptake showed gradual increase in the treatment group. Arrows indicate FDG uptake at each tumor (**A**). Statistical significance was achieved between baseline to 48 h and 24 h to 48 h (**B**). CTRL = control, Mib = mibefradil. * *p* < 0.05.

**Table 1 cancers-15-05244-t001:** Changes in [^3^H] DDG uptake by 72 h mibefradil treatment at different passages.

Time (h) ^†^	[^3^H] DDG Uptake (%) *			Mean ± SE ^‡^
	Exp 1 (Passage 5)	Exp 2 (Passage 6)	Exp 4 (Passage 8)	
0	90.4% (0.569)	71.0% (0.131)	122.5% (0.132)	91.3 ± 12.0%
4	111.3% (0.469)	63.6% (0.110)	110.4% (0.539)	95.1 ± 15.8%
8	104.5% (0.769)	106.1% (0.078)	105.2% (0.407)	105.2 ± 0.5%
12	101.2% (0.902)	100.6% (0.987)	105.8% (0.247)	102.5 ± 1.6%
15	122.8% (0.052)	141.2% (0.279)	110.4% (0.209)	124.8 ± 8.9%
18	134.6% (0.012)	131.8% (0.106)	114.0% (0.048)	126.8 ± 6.4%
24	135.1% (0.001)	147.2% (0.013)	155.1% (0.047)	145.8 ± 5.8%

Numbers in parentheses are the *p*-values for differences in [^3^H] DDG uptake between control and mibefradil-treated cells. * [^3^H] DDG uptake by treated cells was divided by that of control cells at each time point. ^†^ Time in hours after mibefradil removal. ^‡^ Mean ± standard error of relative [^3^H] DDG uptake from three independent experiments. Each experiment was triplicated. [^3^H]-2-DG = [1,2-^3^H(N)]-2-deoxy-d-glucose, Exp = experiment.

## Data Availability

Not applicable.

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
