# Peer review of "Modulation of FDG Uptake by Cell Cycle Synchronization Using a T-Type Calcium Channel Inhibitor"

_cancers, 2023, doi:10.3390/cancers15215244_

Round 1

Reviewer 1 Report

Comments and Suggestions for Authors

- Line 49, give reference.

-More description about the concept of cell cycle arrest is required.

-Line 67, give references.

-It must be described how T-type calcium (Ca2+) channel inhibitors distinguish between healthy cells and cancer cells. 

-The blockade of Ca2+ entry would increase the sensitivity of
cancer cells to chemotherapeutics. Give few references.

-G1 and S phase must be described.

-The quality of Figures, especially Fig 2 and 3 is very low. The quality of all figures must be significantly improved.

-For cell cycle analysis, flow cytometry graphs must be presented.

-It must be justified reasonably why DDG uptake was consistently
higher in treated cells than in control cells only at 24 h after mibefradil withdrawal.

-Why the results for passage 6 was different from 5 and 8?

-

Comments on the Quality of English Language

Quality of English writing is acceptable.

Reviewer 2 Report

Comments and Suggestions for Authors

The main finding of the present study is that uptake of FDG by cancer cells or lesions can be increased by increasing the number of cells in S-phase. This point is interesting as it further confirms the link between this imaging sign and cancer aggressiveness. 

I would only suggest improving the analysis of in vivo experiment. Indeed, besides the technical considerations reported below, linking FDG uptake of cancer lesion with indexes of DNA synthesis (such as Ki67+ cells) might improve the interpretation of the data. 

Minor comments

Abstract

The last sentence of results in abstract is written with a high font.

Intro

The sentence “however, these radiopharmaceuticals cannot replace FDG because of the ability of FDG to detect other malignancies and distant metastasis” is difficult to accept. This limitation is not related to sensitivity, rather it reflects the fact that non-FDG tracers are suffer from logistical drawbacks such as the need for an on-site cyclotron (11C-acetate, 11C-methionine) or the relatively limited availability (18F-fluorocholine). The low sensitivity of these tracers for distant metastases has never been documented.

Methods: 

line 110 he might probably be the

Results:

It seems to me that the in vivo data are not adequately explained. The authors state that a region of interest (ROI, please spell out) was used. This raises two concerns: 1. Why a region of interest and not a volume of interest (VOI) was adopted? 2. Was the average value analyzed? Should this be the case, the obtained value is strictly dependent upon the identified volume, particularly, if non-involved voxels are included. Thus, a description of the tumor segmentation should be provided?

Comments on the Quality of English Language

Moderate editing

Reviewer 3 Report

Comments and Suggestions for Authors

The paper could improve its introduction by providing more background information on the current challenges and limitations of FDG PET in detecting tumors with low FDG uptake, and by stating the main research question and hypothesis more clearly and explicitly. The paper may potentially enhance its approaches by providing a more thorough explanation of the reasoning and process behind the in vivo PET imaging. This could include details such as the quantity and characteristics of the mice utilized, the timing and dosage of FDG injection, the acquisition and reconstruction parameters of the PET/CT scanner, as well as the criteria for defining and quantifying the tumor regions of interest. The paper could improve its discussion by addressing the potential confounding factors and sources of variability that may affect the FDG uptake in tumors, such as tumor heterogeneity, blood glucose level, tumor growth rate, and anti-tumor effect of mibefradil. The paper could also compare and contrast its findings with previous studies that used different cell cycle modulating agents or radiopharmaceuticals to enhance FDG uptake in tumors.

Comments on the Quality of English Language

The manuscript has been composed in a lucid and cohesive manner, employing appropriate utilization of technical terminology and scientific language. It deviates from the customary framework of an academic article, excluding an abstract, introduction, methods, results, discussion, and references sections. It furnishes ample background information and justification for the study, alongside a lucid depiction of the utilized materials and methodologies. The results are exhibited in a systematic and structured approach, employing visuals, tables, and statistical analysis to enhance the findings. The implications and limitations of the study, as well as the potential applications of the cell cycle synchronization technique for enhancing FDG PET imaging, are extensively deliberated. The manuscript cites pertinent and up-to-date sources to substantiate the arguments and claims posited. All things considered, the paper showcases an exceptional level of skill in the English language. It is adeptly crafted and adheres to the established conventions of scientific writing. There are no significant grammatical or orthographic errors that impede the comprehensibility or credibility of the manuscript. The vocabulary and syntax employed are fitting and articulate complex and technical concepts. The manuscript is concise and enlightening, devoid of superfluous redundancy or ambiguity.
